# On the Multidisciplinary Design of a Hybrid Rocket Launcher with a Composite Overwrapped Pressure Vessel

Alain Souza [1,*], Paulo Teixeira Gonçalves [2,*], Frederico Afonso [1], Fernando Lau [1], Nuno Rocha [2] and Afzal Suleman [1,3]

1   IDMEC, Instituto Superior Técnico, Universidade de Lisboa, Av. Rovisco Pais, No. 1, 1049-001 Lisboa, Portugal; frederico.afonso@tecnico.ulisboa.pt (F.A.); lau@tecnico.ulisboa.pt (F.L.); suleman@uvic.ca (A.S.)
2   Institute of Science and Innovation in Mechanical and Industrial Engineering (INEGI), Rua Dr. Roberto Frias, No. 400, 4200-465 Porto, Portugal; nrocha@inegi.up.pt
3   Department of Mechanical Engineering, University of Victoria, Victoria, BC V8W 2Y2, Canada
*   Correspondence: alain.souza@tecnico.ulisboa.pt (A.S.); prgoncalves@inegi.up.pt (P.T.G.)

**Abstract:** A multidisciplinary design optimisation (MDO) study of a hybrid rocket launcher is presented, with a focus on quantifying the impact of using composite overwrapped pressure vessels (COPVs) as the oxidiser tank. The rocket hybrid propulsion system (RHPS) consists of a combination of solid fuel (paraffin) and liquid oxidiser (NOx). The oxidiser is conventionally stored in metallic vessels. Alternative design concepts involving composite-based pressure vessels are explored that could lead to significant improvements in the overall performance of the rocket. This design choice may potentially affect parameters such as total weight, thrust curve, and maximum altitude achieved. With this eventual impact in mind, structural considerations such as wall thickness for the COPV are integrated into an in-house MDO framework to conceptually optimise a hybrid rocket launcher.

**Keywords:** RHSP; MDO; COPV; composites; hybrid propulsion; rockets

## 1. Introduction

The demand for low-cost launch vehicles is expected to increase in the near future due to the increased demand currently not met by the available launch providers [1,2]. Hybrid rocket launchers may provide a sustainable solution by minimising both costs and environmental impact [3,4]. The development of energy-efficient launch vehicle systems and the integration of advanced lightweight materials can increase the affordability of launch vehicles.

Hybrid rocket engines (HRE) are more versatile compared to solid or liquid propellant systems [5]. This versatility is due to the ease in selecting more sustainable fuels that present promise to reduce the carbon footprint of launch vehicles [6,7]. Additionally, HREs are also expected to allow for more affordable access to space based on composite overwrapped pressure vessels (COPVs) [6,7].

In this paper, the performance of a conceptual design of a hybrid propulsion system using a multidisciplinary design optimisation framework is proposed considering the propulsion fuel, the oxidiser storage system, and the rocket structure. The main goal is to study the performance improvement of launchers based on the structural design considerations of a COPV, used as an oxidiser tank. This framework has been verified and validated [8–10] using data collected from different hybrid rockets, namely the Spaceport America® Cup and the European Rocketry Challenge (EuRoC). This choice was made due to the availability of flight data, enabling validation of crucial parameters such as thrust, altitude, and trajectory compared to data from commercial rockets such as Falcon 9, Terran 1, the Long March family, the Ares family, and the Ariane family.

### 1.1. Hybrid Rocket Engines

There are three basic configurations for an HRE [11]: the *classical* or *conventional configuration*, wherein the fuel is in the solid phase and the oxidiser is stored as a liquid; the *inverse* or *reverse configuration*, which has the oxidiser as a solid and the fuel is in liquid form; and the *mixed hybrid configuration*, where a small amount of solid oxidiser is embedded in the solid fuel, and the fuel-rich mixture is then burnt with additional oxidiser injected in an afterburner chamber, or simultaneously at the head and at the end of the grain cavity. Here, the *conventional configuration* is considered. The main characteristics of hybrid propulsion systems are the following [7,11–13]:

- **Advantages**
  1. enhanced protection from explosion or detonation during fabrication, storage, and operation;
  2. start–stop–restart capability;
  3. relative simplicity, which may translate into low overall system cost when compared to liquid bi-propellant engines;
  4. higher specific impulse than solid rocket engines and higher density-specific impulse than liquid bi-propellant engines;
  5. the ability to smoothly change thrust over a wider range of demand.

- **Disadvantages**
  1. mixture ratio and hence specific impulse may vary during steady-state operation (as well as during throttling);
  2. relatively complicated fuel geometries with significant unavoidable fuel residues (slivers) at the end of the burn, which somewhat reduces the mass fraction and can vary if there is random throttling;
  3. prone to large-amplitude low-frequency pressure fluctuations (termed chugging);
  4. relatively complicated internal motor ballistics resulting in incomplete descriptions, both of regression rates of the fuel and of scaling effects, affecting the design of large hybrid systems.

The HRE with a *typical configuration* is constituted in [7,14]: a pressure tank filled with an inert gas, the oxidiser tank, a valve responsible for controlling the oxidiser flux within the combustion chamber, an injector to create a diffuse oxidiser spray, an ignitor to start the reaction, the combustion chamber where the fuel grain is located, and a nozzle. Figure 1 shows in more detail the configuration of the HRE.

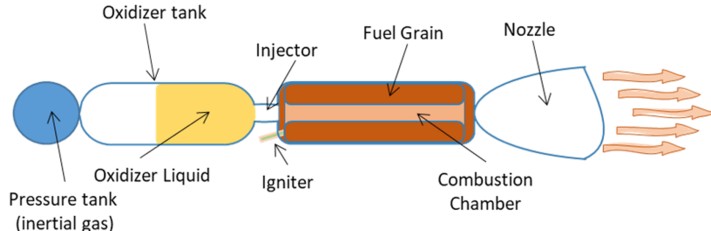

**Figure 1.** Conceptual diagram of an HRE.

It is possible to reduce the mass of the propulsion system by using self-pressurising oxidising gas, such as $N_2O$ or LOX, for example. In this case, the need for a separate tank with inert gas can be eliminated. In the current study, a combination of nitrous oxide ($N_2O$) oxidiser and paraffin-based fuel is used. To have self-pressurising properties, the $N_2O$ needs to be stored at a pressure of 50 bar, requiring a robust vessel construction such as a COPV to minimise the increase in dry mass in the overall concept of the rocket [15].

### 1.2. Composite Overwrapped Pressure Vessels

The COPVs shown in Figure 2 range from type I vessels to type V vessels, where type V vessels can also be found as linerless COPVs. Composite materials are attractive

to be used as reinforcement systems because they exhibit a high strength-to-weight ratio, which is crucial in launchers and other space applications.

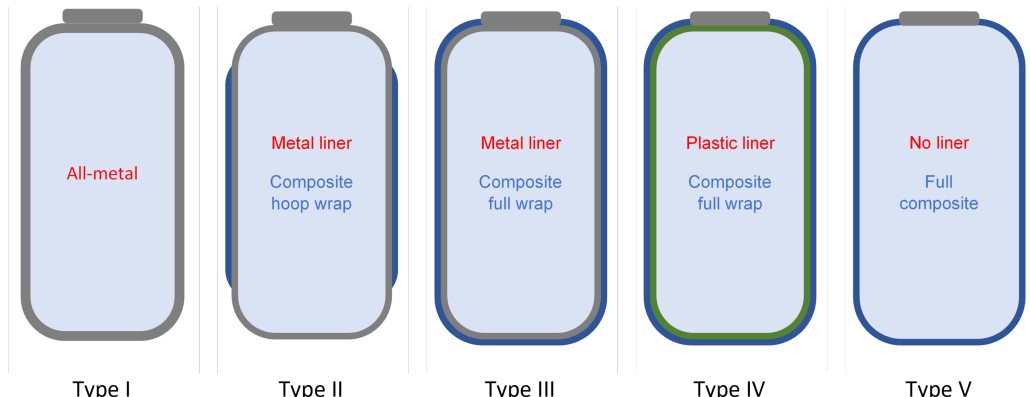

**Figure 2.** Pressure vessel types according to their construction.

Historically, type I has had a market share of more than 90% because it is cheaper and the most cost-effective solution for the oil industry. Type III and type IV vessels use composites to reduce weight and increase compressed gas storage efficiency in the aerospace industry [16,17]. Type V vessels have great potential to further increase storage efficiency in the transportation and aerospace industries but are still in development and present several design and production challenges to industrialise [18,19] and incorporate into the market. However, there are additional challenges for type V COPVs to be used in the storage of liquid cryogenic substances [20,21]. Cryogenic storage offers additional advantages because gases such as oxygen and methane can be stored as liquids, reducing the required volume and storage pressure for the same mass, but more difficulties arise because of the operation temperature. COPV modelling requires appropriate consideration of the geometrical features of the configuration. Different works can be found in the literature dealing with winding process modelling and thickness profile prediction [22–26], which are crucial to generate a high-fidelity geometric model of COPV with the appropriate path profiles.

However, constitutive modelling of FRP requires appropriate consideration to address the ortotropic material response. Actually, the mechanical response is transversely isotropic, where the direction of the fibre is located along the principal axis. The mechanical properties of the fibre usually outperform the properties of the polymer matrix system and dominate the response of the material in the longitudinal direction, while the transverse direction is significantly affected by the matrix and interface properties [27]. The first failure criteria for FRP were based on 2D formulations [28]. The Hashin failure model has been extensively used to analyse the failure of unidirectional composite materials with two failure mechanisms: fibre failure or longitudinal failure and matrix failure or transverse failure.

After the world-wide failure exercise, Puck's criteria [29] gained momentum, emphasising that transverse failure is developed in a plane parallel to the fibres and that the orientation of this plane depends on the stress state. Since then, different constitutive models have been developed to simulate failure initiation and damage evolution, most of them using a continuum damage approach employing the crack band theory [30,31].

Considering the potential weight reductions with the incorporation of full-composite vessels for the oxidant tank, the incorporation of the linerless tank concept into the MDO architecture will enable further optimisation of the rocket design.

## 2. Modelling Approaches

The mathematical modelling of the MDO approach is presented. This framework consists of three disciplines, propulsion, mass and size, and aerodynamics, as illustrated in Figure 3. Each one of these subjects will be covered in a separate sub-section. Additionally,

the COPV oxidiser tank modelling included in the mass and sizing discipline will be present as well.

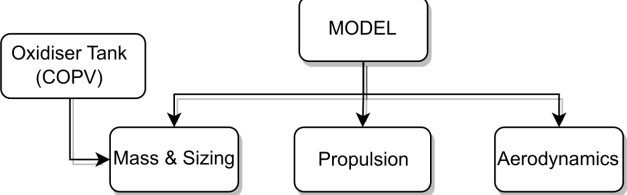

**Figure 3.** Structure of the disciplines.

### 2.1. Propulsion Modelling

The main considerations adopted for the propulsion model are provided in Figure 4.

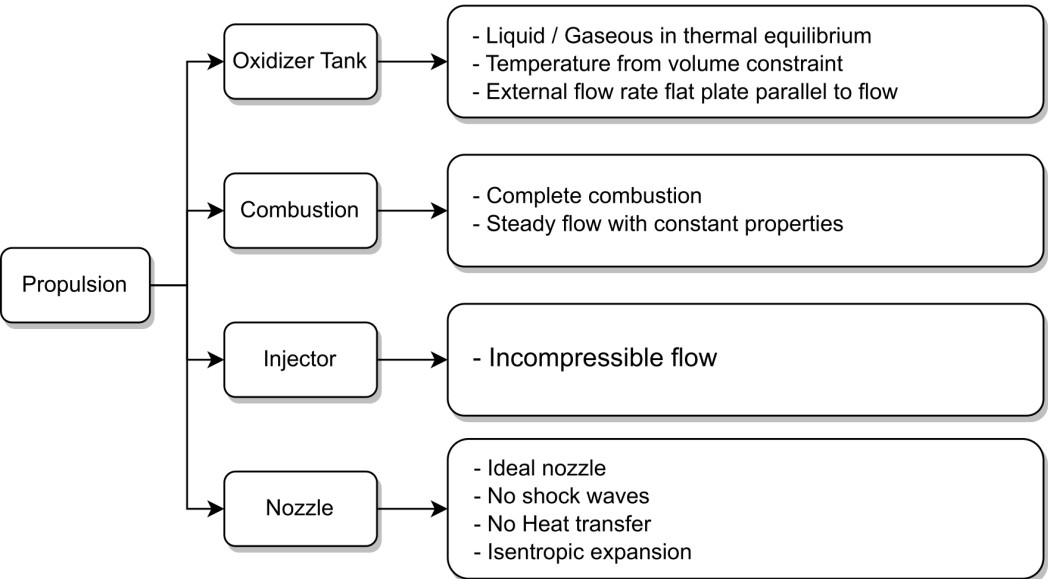

**Figure 4.** Propulsion assumptions.

This model describes the iterations between the oxidiser and the fuel to develop the theoretical model of the fuel combustion process. The assumptions include complete and instantaneous combustion processes, incompressible single-phase fluid flow, and thermal equilibrium between the liquid and gas phases in saturated form [32–34]. The thermodynamic properties required for the model are extracted from the National Institute for Standards and Technology (NIST) [15].

With the information on the saturated conditions of the oxidiser from its melting point up to the critical temperature, it is possible to establish the following relationship:

$$p_{OT} - p_{feed} - p_{cc} > 0 \, , \tag{1}$$

where, with oxidiser vapor pressure (which corresponds to the tank pressure, $p_{OT}$) and the pressure loss through the injector and feed system, $p_{feed}$, the possibility of oxidiser flow to the combustion chamber $p_{cc}$ is verified.

For the oxidiser mass flow rate $\dot{m}_{ox}$,

$$\dot{m}_{ox} = C_{inj} \sqrt{2\rho_d (p_{ot} - p_{feed} - p_{cc})} \, , \tag{2}$$

where $\rho_d$ is the discharge fluid density and $C_{inj}$ is the effective injection area, this parameter is obtained by multiplying the injector area by a discharge coefficient.

Assuming that the fuel grain has a cylindrical shape, the regression rate (Figure 5, shows the radial direction of the fuel burning) can be expressed in the generic form provided by

$$\dot{r} = aG_o^n \, ,$$ (3)

where the parameters $a$ and $n$ are empirically fitted and are highly dependent on the propellant choice, while $G_o$ represents the ratio of the oxidiser mass flow rate to the port section area, gradually increasing as the fuel is consumed.

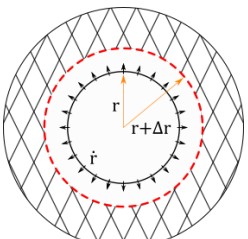

**Figure 5.** Fuel regression diagram.The arrows indicate the direction of the fuel burning.

The initial value of $G_o$ is considered as a ratio between the oxidiser mass flow rate and the port area (4):

$$G_o = \frac{\dot{m}_{ox}}{\pi r_{port}^2} \, .$$ (4)

The mass fuel rate $\dot{m}_f$ is calculated as a function of the regression rate $\dot{r}$, fuel length $L_f$, fuel density $\rho_f$, and port radius $r_{port}$:

$$\dot{m}_f = 2\pi r_{port} \dot{r} L_f \rho_f \, .$$ (5)

Then, with the oxidiser and fuel mass rate, it is possible to calculate the total mass flow $\dot{m}_{cc}$:

$$\dot{m}_{cc} = \dot{m}_f + \dot{m}_{ox} \, .$$ (6)

The next iteration of $G_{i+1}$ is provided by

$$G_{i+1} = \frac{\dot{m}_{ox,\,i} + \dot{m}_{cc,\,i}}{2\pi r_{port}^2} \, .$$ (7)

Admitting an ideal nozzle (isentropic, without shock waves, steady axial flow, and homogeneous fluid), the stagnation pressure $p_0$ can be calculated as a function of stagnation temperature $T_0$, fluid heat capacity ratio $k$, and ideal gas constant $R$:

$$p_o = \frac{\dot{m}_{cc}}{\zeta_d A_{th}} \sqrt{\frac{T_o R}{k} \left( \frac{k+1}{2} \right)^{\frac{k+1}{k+2}}} \, .$$ (8)

In Equation (8), $\zeta_d$ is a discharge correction factor associated with model imperfections and $A_{th}$ is the throat area of the nozzle. The pressure of the combustion chamber $p_{cc}$ is calculated as follows:

$$p_{cc} = p_0 \left( \frac{T_{cc}}{T_0} \right)^{\frac{k}{k-1}} \, ,$$ (9)

where $T_{cc}$ is the combustion chamber temperature.

Using NASA software *Chemical Equilibrium with Applications* (CEA) (LEW-17687-1)[35], it is possible to obtain the stagnation temperature $T_0$, specific heats $c_{p,cc}$ and $k_{cc}$, and density $\rho_{cc}$, with the combustion chamber pressure $p_{cc}$ and the propellant ratio,

$$O/F = \frac{\dot{m}_{ox}}{\dot{m}_f} \, .$$ (10)

To determine the combustion chamber temperature, the following expression is employed:

$$T_{cc} = T_0 - \frac{v_{cc}^2}{2c_{p,cc}} \, , \tag{11}$$

where $v_{cc}$ is the particles velocity,

$$v_{cc} = \frac{\dot{m}_{cc}}{\rho_{cc} A_{cc}}, \tag{12}$$

$c_p$ is the specific heat, and $A_{cc}$ is the combustion chamber area.

To take into account neglected effects, such as heat transfer losses and incomplete combustion, a correction factor can be applied for combustion efficiency, denoted by $\zeta_c$. This factor can be used to modify the theoretical stagnation temperature using CEA software [35].

It is possible to calculate the Mach number ($M_e$) iteratively, relating the nozzle throat area $A_{th}$ to the nozzle exit area $A_e$ with the following equation,

$$\left( \frac{A_e}{A_{th}} \right)^2 = \frac{1}{M_e^2} \left[ \frac{2}{k+1} \left( 1 + M_e^2 \frac{k-1}{2} \right) \right]^{\frac{k+1}{k-1}} . \tag{13}$$

As the liquid oxidiser in the tank is depleted, a drop in the pressure of the combustion chamber can cause the outlet pressure to drop, potentially leading to separation of the outlet flow. Then, with the results of Equation (13), it is possible to obtain the velocity ($v_e$), pressure ($p_e$), and temperature ($T_e$) at the nozzle exit as follows:

$$T_e = T_0 \left( 1 + M_e^2 \frac{k-1}{2} \right)^{-1} , \tag{14}$$

$$p_e = p_0 \left( 1 + M_e^2 \frac{k-1}{2} \right)^{-\frac{k}{k-1}} , \tag{15}$$

$$v_e = M_e \sqrt{kRt_e} . \tag{16}$$

The thrust $F_{t,prop}$ generated by the propulsion system is calculated as a function of atmospheric pressure $p_{atm}$, the exit nozzle area $A_e$, the exhaust mass flow $\dot{m}_e = \dot{m}_{cc}$, the exit velocity $v_e$, and upstream stagnation properties, as expressed.

$$F_{t,prop} = \zeta_{Cf}(\dot{m}_{cc} v_e + (p_e - p_{atm}) A_e) , \tag{17}$$

where the parameter $\zeta_{Cf}$ is a correction factor not accounted for when using an ideal nozzle model.

The thermal resistance $R_{conv,in}$, due to internal convection, is modelled by admitting a cylindrical one-dimensional wall and without internal energy generation,

$$R_{conv,in} = \frac{\ln (r_{cc}/r_{OT})}{2\pi L_{OT} k_{eff}} . \tag{18}$$

In Equation (18), the thermal resistance is defined as a function of the oxidiser tank geometry (length and radius) ($L_{OT}$, $r_{OT}$), the effective thermal conduction ($k_{eff}$), and combustion chamber radius ($r_{cc}$).

The thermal resistance $R_{cond}$, due to the conduction through the external structure, is defined as a function of the external thermal conductive $k_{ex}$, the radius of the combustion chamber $r_{cc}$, and the external radius $r_{ex}$,

$$R_{cond} = \frac{\ln (r_{ex}/r_{cc})}{2\pi L_{OT} k_{ex}}. \tag{19}$$

In the end, the external convection model is provided using the simplified model of a flat plate parallel to the flow of length $2\pi r_e x$. Then, the thermal resistance $R_{conv,ex}$ is

$$R_{conv,ex} = \frac{1/\bar{h}_c}{2\pi L_{OT} r_{ex}} \,,$$
(20)

where $\bar{h}_c$ is the average convection coefficient. With Equations (18)–(20), the heat transferred $Q$ is calculated as

$$Q = \frac{T_{OT} - T_{amb}}{R_{conv,in} + R_{conv,ex} + R_{cond}},$$
(21)

where the parameters $T_{OT}$ and $T_{amb}$ are the oxidiser tank and ambient temperatures (Equation (21)).

### 2.2. Mass and Sizing

The main considerations adopted for the mass and sizing model are provided in Figure 6.

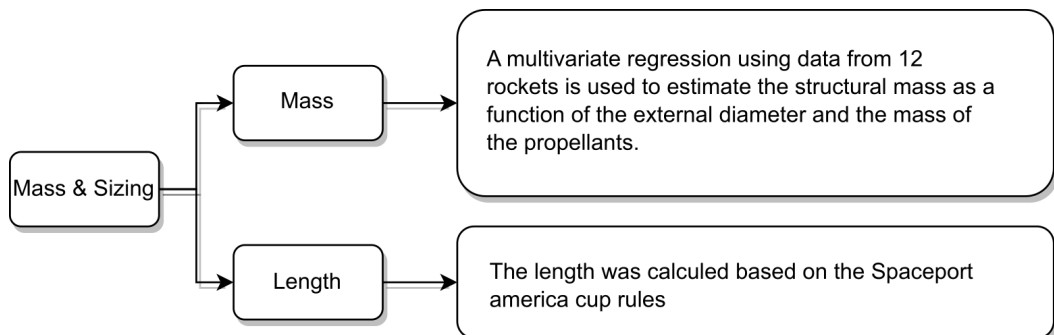

**Figure 6.** Mass and sizing assumptions.

In the conceptual design phase, it is difficult to accurately estimate the total mass and dimensions of a rocket. For this reason, these values are estimated using linear regressions based on the available data [8–10,32]. A collection of rockets from the 2019 edition of the Spaceport America Cup (SAC) are used as a dataset [9].

With that information, the regression expression for the structural mass $m_{strc}$ as a function of the propellant mass $m_{prop}$ ($m_{prop} = m_{ox} + m_f$) and the external diameter $D_{ex}$ is provided by

$$m_{strc} = 0.45\, m_{prop} + 259.01\, D_{ex} - 13.26 \,.$$
(22)

The calculated structural mass represents the sum of the masses of all components of the rocket, including the mass of the oxidiser tank.

Two regressions were conducted to determine the size of certain components. The first regression aimed to establish the length of the recovery bay ($L_{rec}$) based on the total length of the rocket ($L_{rocket}$),

$$L_{rec} = 0.24\, L_{rocket} - 0.42 \,.$$
(23)

The second regression aimed to estimate the length of the avionics bay ($L_{av}$) based on the length of the recovery bay,

$$L_{av} = 0.69\, L_{rec} - 0.078 \,.$$
(24)

The payload bay has a constant length of $L_{pay} = 0.4$ m, for all rockets considered in the dataset, due to one of the requirements of the SAC. The lengths of the nose cone ($L_{cone}$) and the nozzle ($L_{nozzle}$) are independent project variables that can vary according to manufacturing costs. These variables pose challenges in associating them with other

known parameters due to their individual nature and the flexibility in determining their values. Then, initial values of 0.6 m and 0.14 m are assumed, respectively.

### 2.3. Aerodynamics and Stability Modelling

The main considerations adopted for the aerodynamics and stability model are provided in Figure 7.

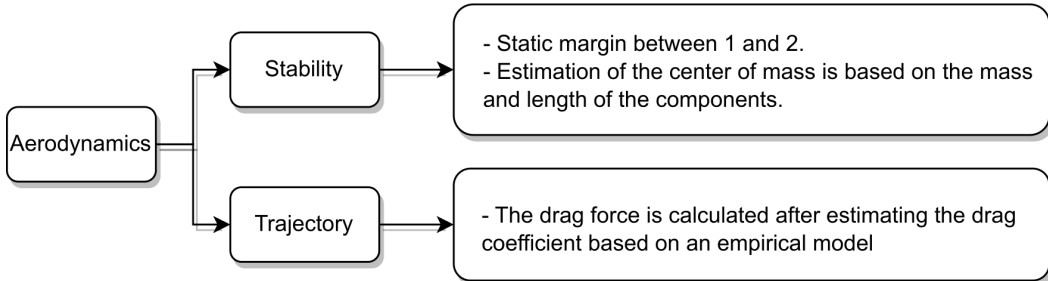

**Figure 7.** Aerodynamic and stability assumptions.

In the analysis of aerodynamics and stability, it is necessary to determine the pressure centres ($CP$) and gravity centres ($CG$) of the rocket. To determine $CG$, a weighted average is created with respect to the centre of gravity position $d_i$ and its mass $m_i$, as expressed in the following equation.

$$CG = \frac{\sum d_i m_i}{\sum m_i} \ . \tag{25}$$

The $CP$ is obtained by the weighted sum of the normal force coefficients of the components $(C_N)_i$ and the centre of pressure ($CP_i$),

$$CP = \frac{\sum (C_N)_i CP_i}{\sum (C_N)_i} \ . \tag{26}$$

Body and nose contributions are assumed to be included in the normal force coefficient $(C_N)_i$ and are assumed to be equal to $(C_N)_{cone} = 2$, and the centre of pressure is located

$$CP_{cone} = \frac{2}{3} L_{cone} \ , \tag{27}$$

where $L_{cone}$ is the distance from nose tip until the nose centre. Now, $(C_N)_{fins}$ is calculated as a function of the radius of the rocket $R$, the semi-span of the fin $S$, the diameter of the nose $d$, the length of the mid-chord $L_F$, the root chord of the fin $C_R$, the tip chord $C_T$, and the number of fins $N$:

$$(C_N)_{fins} = \left[1 + \frac{R}{S + R}\right] \left[\frac{4N(S/d)^2}{1 + \sqrt{1 + [(2L_F)/(C_R + C_T)]^2}}\right] \ . \tag{28}$$

The centre of pressure $CP_F$ relative to the fins is provided by

$$CP_F = x_B + \frac{x_R(C_R + 2C_T)}{3(C_R + C_T)} + \frac{1}{6}\left[(C_R + C_T) - \frac{C_R C_T}{C_R + C_T}\right] \ , \tag{29}$$

where $x_B$ is the leading edge of the chord and $x_R$ is the length between the leading edge of the fin root and the body.

### 2.4. Tank Modelling

One of the critical components of the rocket design is the oxidiser tank. The reference operating pressure is 50 bar, combined with temperature changes and flight dynamic loads. At this stage, a preliminary conceptual design will be developed, considering

internal pressure as the principal design requirement, to obtain the design layup and mass properties that will serve as input to the MDO model.

The geometry of the tank corresponds to a cylindrical vessel with spherical heads, an external diameter of $D_{ext} = 100$ mm, and length $L_{tank} = 200$ mm, as shown in Figure 8.

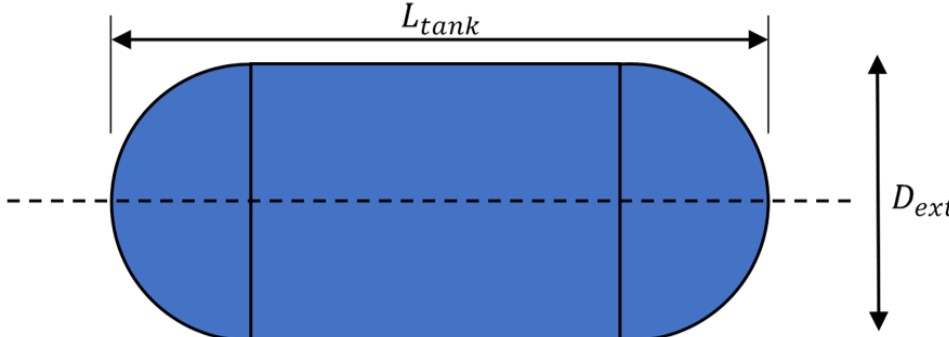

**Figure 8.** Vessel geometry.

Therefore, to analyse the mechanical behaviour of the oxidant pressure vessel, a finite element (FEM) model was developed, to support the composite layup design, considering the layer thickness using Wang's model [22], and 3D composite failure criteria [30,36]. To perform the model verification, the unidirectional carbon fibre/epoxy system IM7/8552 is used because it is commonly applied in this application and the material properties are widely documented in the literature. Therefore, Table 1 presents the elastic properties, and Table 2 presents the failure stresses from [37]. The ply thickness is 0.131 mm.

**Table 1.** Elastic properties [37].

| Symbol | Description | Value | Units |
|---|---|---|---|
| $E_L$ | Longitudinal elastic modulus | 171,420 | MPa |
| $E_T$ | Transversal elastic modulus | 9080 | MPa |
| $G_{LT}$ | Longitudinal shear modulus | 5290 | MPa |
| $v_{LT}$ | Longitudinal Poisson's ratio | 0.32 | n/d |
| $v_{TT}$ | Transverse Poisson's ratio | 0.4 | n/d |

**Table 2.** Strength properties [37].

| Symbol | Description | Value | Units |
|---|---|---|---|
| $X_T$ | Longitudinal uniaxial tension | 2323.5 | MPa |
| $X_C$ | Longitudinal uniaxial compression | 1200 | MPa |
| $Y_T$ | Transverse uniaxial tension | 62.3 | MPa |
| $Y_C$ | Transverse uniaxial compression | 253.7 | MPa |
| $S_L$ | Longitudinal shear | 89.6 | MPa |
| $S_T$ | Transverse shear | 81.1 | MPa |
| $Y_{BT}$ | Transverse biaxial tension | 38.7 | MPa |
| $Y_{BC}$ | Transverse biaxial compression | 3501 | MPa |

The FEM model was developed using Abaqus ®; the geometric model corresponds to 1/8 of the complete vessel to reduce computational effort, as shown in Figure 9. The model mesh corresponds to one solid element (C3D8R) per layer with a size of 0.5 mm (from mesh convergence analyses); the element thickness corresponds to the layer thickness.

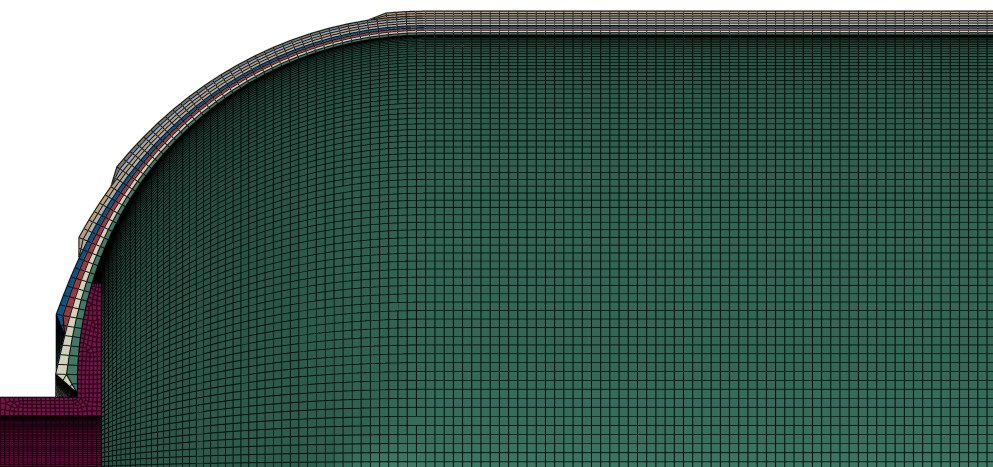

**Figure 9.** COPV finite element mesh.

The layer thickness is estimated using Equation (30) from Wang's model [22], and the fibre trajectories correspond to geodesic paths, which are calculated using Equation (31) from [23], where $t_{ck_i}$ is the ply thickness at the point $i$, $R_0$ is the vessel radius in the cylindrical section, $r_i$ is the radius at point $i$ in the dome region, $t_{ck_0}$ is the ply nominal thickness, $z$ is the longitudinal position coordinate, and $\theta$ is the fiber direction measured from the axis as shown in Figure 10.

$$t_{ck_i} = \frac{R_0}{r_i} t_{ck_0} . \tag{30}$$

$$\frac{d\theta}{dr} = \frac{dr}{dz} \frac{tan(\theta)}{r_i} . \tag{31}$$

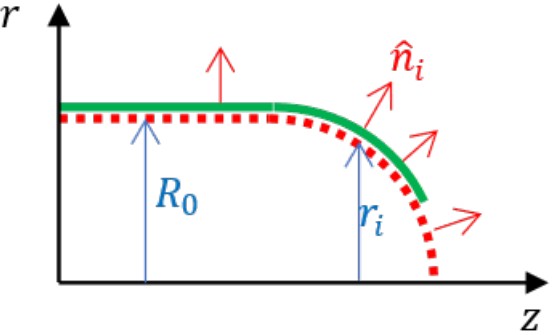

**Figure 10.** Path geometric description.

The material for the external fitting is aluminium with an elastic modulus of $E_{al} = 72$ GPa and a Poisson ratio of 0.3. The use of 3D solid elements enables using the full-3D-based failure criteria. The connection between the aluminium fitting and the composite was modelled using a cohesive interface with a stiffness $K = 10^6$ N/mm. Interface failure was not considered in this analysis [38].

A preliminary design of the COPV layup is performed using classic laminate theory using the first ply failure criteria and ensuring full coverage of the vessel dome. The laminate is defined in $\pm\theta$ layers to be adequate for filament winding or equivalent processes for axis-symmetric geometries. The results of the preliminary COPV design and posterior simulations suggest a laminate with the following configuration: $[\pm15_2, \pm30_2, \pm45_2, \pm60_2, \pm90_4]$. Figure 11 shows the deformed tank under the internal pressure of 50 bar; Figure 12 shows the von Mises stress distribution at the metallic head.

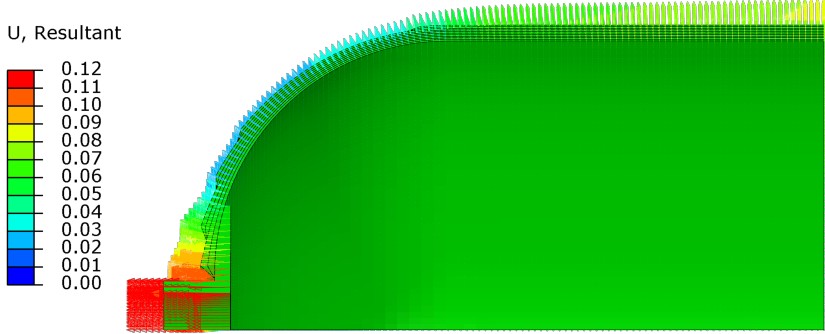

**Figure 11.** Displacement field of the COPV model.

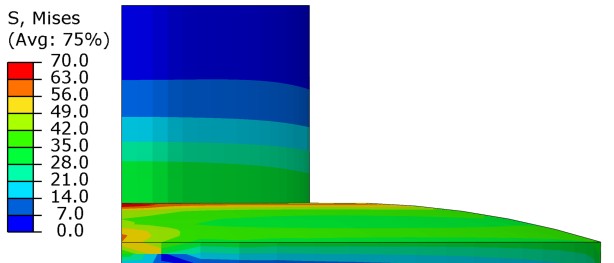

**Figure 12.** The von Mises stress distribution for metallic head.

Figure 13 shows the fiber stress distribution throughout the geometry of the vessel for the same internal pressure of 50 bar.

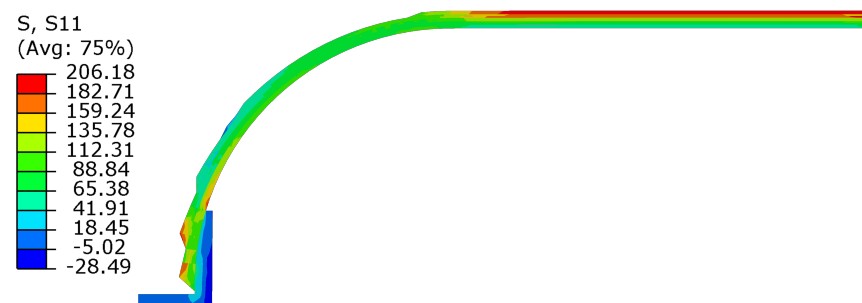

**Figure 13.** Fibre principal direction stress distribution.

However, for the unidirectional fibre material, additional failure criteria are required to appropriately evaluate the proposed design by means of a longitudinal failure index $r_L$

$$r_L = \epsilon_{11} \frac{E_L}{X_T} \, , \tag{32}$$

which relates the stress state in the fibre direction $S_{11}$ with the failure stress and corresponds to the solution-dependent variable SDV29, and by transverse failure index $r_T$

$$r_T = \delta_1 I_1 + \delta_2 I_2 + \delta_3 I_3 + \delta_{32} I_3^2 \, , \tag{33}$$

which relates the transverse stress state regarding $S_{22}$, $S_{33}$, $S_{12}$, $S_{13}$, $S_{23}$ using stress-invariant measures $I_1$, $I_2$, $I_3$ to the matrix-dominated failure strengths $Y_T$, $Y_C$, $S_L$, $S_T$, $Y_B T$, $Y_{BC}$ that are used to calculate the failure envelope parameters $\delta_1, \delta_2, \delta_3, \delta_{32}$. For further details, please refer to [31]. The failure indicators have a homogeneous axis-symmetric distribution because the actual COPV laminate is also symmetric; hence, slice cut of the results is presented in Figures 14 and 15.

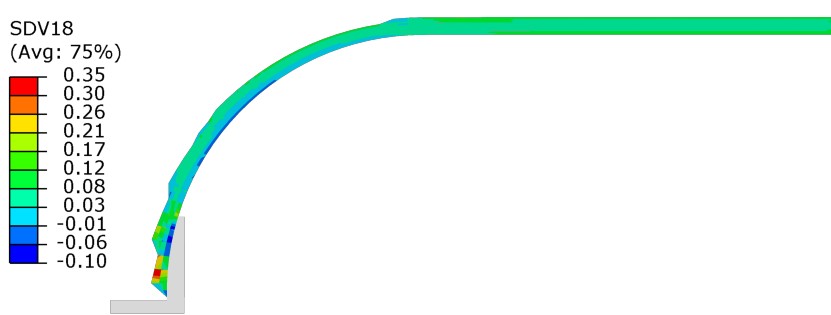

**Figure 14.** Transverse failure criteria $r_T$ (SDV18).

The failure indicator suggests that the failure criteria for fibre tensile rupture are below 10% and for transverse failure are below 20%. This design represents a conservative approach because additional loading cases regarding flight conditions like thrust, vibrations, aerodynamic loads, thermal strains, etc., are not considered in this preliminary evaluation. Transverse failure corresponds to the weakest failure mode, which could lead to fluid leakage. Internal pressure is clearly the most relevant load, however, but thermal strain could also have a severe impact on transverse loads, which should be considered in a more detailed analysis because they could achieve near 50% of the load carrying capacity of the material [39]. This COPV design has a total weight of 0.262 kg, where each head has 0.057 kg and the cylindrical portion has 0.148 kg, which corresponds to 1.48 kg/m.

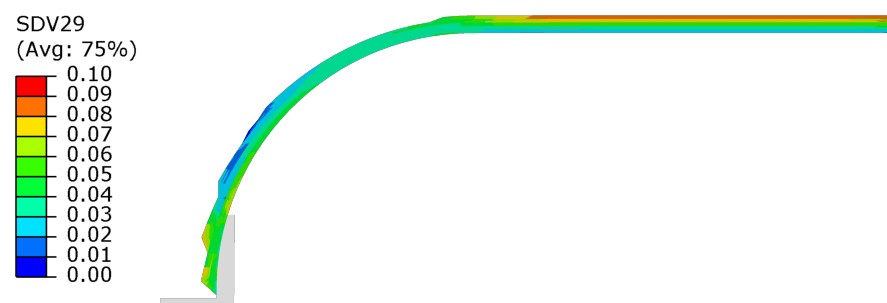

**Figure 15.** Longitudinal failure criteria $r_L$ (SDV29).

## 3. Multidisciplinary Design Optimisation

MDO refers to a framework or approach that integrates multiple engineering disciplines and their associated models and algorithms into a single optimisation problem [40]. The goal of MDO is to find the best design solution that satisfies all the requirements and constraints of different disciplines, such as aerodynamics, structures, propulsion, and controls. By considering the interactions and dependencies between these disciplines, MDO aims to improve the overall performance and efficiency of the design.

This problem involves the engineering disciplines, aerodynamics, stability, propulsion, and mass and sizing proposed by [8–10],

The main objectives of this optimisation are to optimise the rocket-specific impulse $I_{sp}$ and minimise the total mass $m_{rocket}$; then, the functional $J(x)$ to be optimised is a weighted sum provided by

$$J(x) = \sum_{i}^{n} w_i f_i(x) , \tag{34}$$

where $w_i$ represents the weight, with a value ranging from 0 to 1, and the sum of all weights $n$ equals one. $f_i$ denotes a function that correlates the objectives of interest [41]. In the present analysis, we consider $n = 2$, and the functional $J(x)$ assumes the following value:

$$J(x) = -w I_{sp} + (1-w) m_{rocket} . \tag{35}$$

where, in Equation (35), $I_{sp}$ is the specific impulse, $m_{rocket}$ is the total mass, and $w$ is a weight penalisation.

For the weight $w$, a tradeoff must be analysed. Higher values will assign greater importance to the specific impulse, resulting in improved engine performance but potentially leading to a heavier vehicle. On the other hand, lower values of $w$ tend to achieve the opposite effect.

### 3.1. MDO Architecture

The MDO architecture refers to the specific manner in which the optimisation problem is structured and solved. In this study, the Individual Discipline Feasible (IDF) approach was chosen. The IDF approach offers the advantage of solving disciplines independently from each other in a given iteration [41]. This means that completely dissimilar systems can be modelled at a given time before the optimisation algorithm ensures convergence of the equality constraints.

$$
\begin{aligned}
\text{Maximize} \quad & I_{sp}(\mathbf{x}, \mathbf{y}(\mathbf{x}, \hat{\mathbf{y}})) \\
\text{with respect to} \quad & \mathbf{x}, \hat{\mathbf{y}} \\
\text{subject to} \quad & c_0(\mathbf{x}, \mathbf{y}(\mathbf{x}, \hat{\mathbf{y}})) \leq 0 \\
& c_i(\mathbf{x}_0, \mathbf{x}_i, \mathbf{y}_i(\mathbf{x}_0, \mathbf{x}_i, \hat{\mathbf{y}}_{j \neq i})) \leq 0 \quad \text{for } i = 1, \dots, N \\
& c_i^c = \hat{\mathbf{y}}_i - \mathbf{y}_i(\mathbf{x}_0, \mathbf{x}_i, j \hat{\neq} i) = 0 \quad \text{for } i = 1, \dots, N
\end{aligned}
\tag{36}
$$

The Extended Design Structure Matrix (XDSM), proposed by Lambe and Martins [42], is a type of diagram used to visualise MDO architectures, illustrating the interdependence of disciplines and the flow of processes. It offers a convenient way to depict relationships and interactions within an MDO framework. The IDF architecture of the rocket MDO is expressed in an XDSM chart in Figure 16.

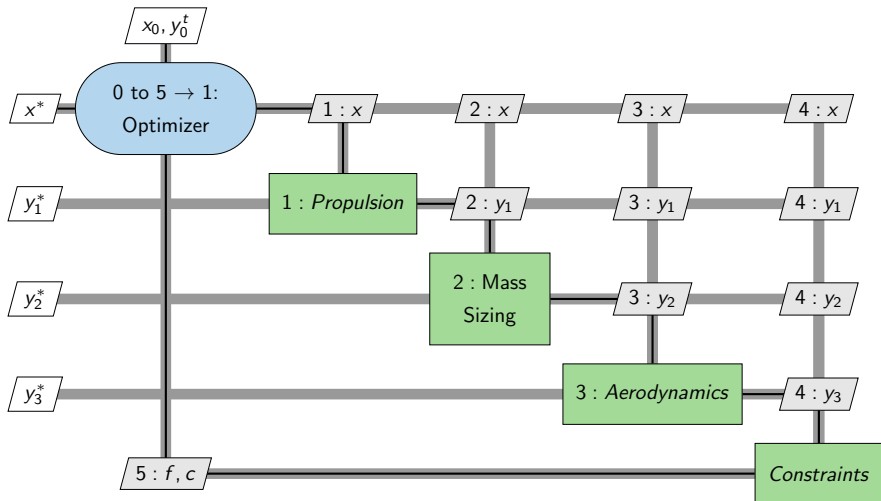

**Figure 16.** IDF architecture in an XDSM chart.

In simple terms, the components are executed in a particular order to ensure convergence. They exchange information and follow a specific flow of data. The inputs are represented by vertical lines, the outputs are represented by horizontal lines, and the data flow follows a clockwise direction. The disciplines are based on certain state variables obtained from the analysis of other disciplines in previous steps.

### 3.2. Constraints

To ensure physical feasibility, certain constraints are introduced in the optimisation setup. For this study, the inequality and equality constraints are considered.

The first inequality constraint is correlated with the pressure limits in the combustion chamber $p_{cc}$, which are limited by an 80% difference between the pressure of the oxidiser tank $p_{ot}$ and the pressure drops of the feed system $p_{feed}$, as follows.

$$p_{cc} - 0.8(p_{ot} - p_{feed}) \; < \; 0 \,. \tag{37}$$

Then, the combustion chamber pressure is limited to 80% of the difference between the pressure of the oxidiser tank $p_{ot}$ and the pressure of the feed system $p_{feed}$.

To avoid combustion instability, the mass flow $G$ is limited to

$$G - 500 \; < 0 \,. \tag{38}$$

Regarding rocket stability, this is guaranteed by the static margin $SM$,

$$1 - SM < \; 0 \,. \tag{39}$$

The dimension restriction of

$$r_{ot} - r_{cc} < \; 0 \tag{40}$$

is to avoid that the oxidiser radius $r_{ot}$ is larger than the combustion chamber radius $r_{cc}$. To limit the length of the grain fuel $L_{fuel}$, this parameter should be lower than the difference between the length of the combustion chamber $L_{cc}$ and the external diameter of the rocket $D_{ex}$, as follows.

$$L_{fuel} + D_{ex} - L_{cc} < \; 0 \,. \tag{41}$$

Then, the fin tip chord $B_{fin}$ cannot exceed 90% of the root chord $b_{fin}$,

$$b_{fin} - 0.9B_{fin} < \; 0 \,. \tag{42}$$

### 3.3. Optimisation Algorithm and Configuration

To solve this MDO problem, a MATLAB $^\circledR$ script was developed. The script utilises the *fmincon* solver, employing the interior-point algorithm [41] and incorporating a multipleStart solver.

The decision to use the MultiStart approach was driven by the fact that the system has 32 design variables, making it highly constrained. By employing a MultiStart Random Start Point Set, the script can efficiently explore various starting points and find feasible solutions more effectively. This approach increases the chances of obtaining optimal or near-optimal solutions, especially for complex and highly constrained problems like this one.

### 4. Results and Discussion

To investigate the impact of the oxidiser tank on the rocket design, the previously described MDO framework was employed. This study aims to analyse how the performance and mass of the rocket change based on the thickness of the tank. Figure 17 summarises the software architecture, taking into account the thickness value update.

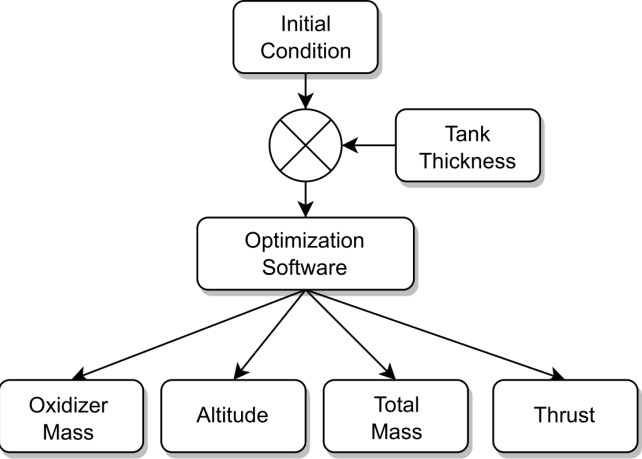

**Figure 17.** Software architecture.

To maintain validation of the simulation software [8–10], the design is based on a hybrid combustion-powered competition rocket with the aim of achieving the lowest possible weight for a minimum target altitude of 3000 m.

In the optimisation analyses, six values were selected for the oxidiser tank thickness: 9 mm, 8 mm, 7 mm, 6 mm, 5 mm, and 4 mm. To enable fair comparison, in the optimisation model, two designs are considered for the oxidiser tank: one constructed with aluminium ($\rho = 2700$ kg/m$^3$) and the other with COPV ($\rho = 1400$ kg/m$^3$).

The initial values of the 32 design variables are presented in Table 3 alongside their corresponding lower and upper boundaries.

**Table 3.** The initial conditions for the optimisation.

| Variable | Parameter | Values | Units |
|---|---|---|---|
| $V_{tank}$ | oxidiser tank volume | 0.012 | m$^3$ |
| $C_{inj}$ | effective injector area | 2.100E-05 | m$^2$ |
| $L_f$ | fuel grain length | 0.200 | m |
| $d_{port}$ | initial fuel grain port diameter | 0.080 | m |
| $d_{th}$ | nozzle throat diameter | 0.024 | m |
| $A_{ratio}$ | nozzle area ratio as a fraction | 3.000 | - |
| $M_{rocket}$ | rocket's mass | 35.12 | kg |
| $L_{tube}$ | structure length | 3.342 | m |
| $D_{ex}$ | rocket's external diameter | 0.100 | m |
| $m_{cc}$ | combustion chamber mass | 1.529 | kg |
| $m_{OT}$ | oxidiser tank mass | 3.000 | kg |
| $M_{tube}$ | external structure mass | 6.000 | kg |
| $L_{OT}$ | oxidiser tank length | 1.557 | m |
| $L_{cc}$ | combustion chamber length | 0.500 | m |
| $D_{fin}$ | fin span | 0.059 | m |
| $r_{OT}$ | oxidiser tank radius | 0.051 | m |
| $B_{fin}$ | fin root chord | 0.300 | m |
| $b_{fin}$ | fin tip chord | 0.225 | m |
| $L_{rec}$ | recovery system length | 0.554 | m |
| $L_{av}$ | avionics system length | 0.303 | m |
| $m_{av}$ | avionics system mass | 3.241 | kg |
| $m_{rec}$ | recovery system mass | 7.778 | kg |
| $m_{fins}$ | fins mass | 0.199 | kg |
| $m_{nozzle}$ | nozzle mass | 1.944 | kg |
| $m_{cone}$ | cone mass | 0.817 | kg |
| $d_{OT.i}$ | distance between nose cone tip and oxidiser tank | 1.858 | m |
| $r_{ccin}$ | combustion chamber internal radius | 0.055 | m |
| $m_f$ | fuel mass | 0.855 | kg |
| $m_{ox}$ | oxidiser mass | 6.835 | kg |
| $r_{cc}$ | combustion chamber radius | 0.058 | m |
| $m_{add}$ | additional mass in the recovery bay | 1.500 | kg |
| $m_{add.cone}$ | additional mass in the nose cone | 0.000 | kg |

Figure 18a,b show the maximum values achieved for altitude and thrust, respectively, after the optimisation process as functions of the thickness of the oxidiser tank. In Figure 18b, the highest thrust value for the case of the aluminium tank is 3180.82 N with a thickness of 7 mm and for the COPV tank 2387.07 N with a thickness of 5 mm. In terms of the average for the aluminium tank, the maximum thrust is 2760.51 N and for the COPV tank is 2102.57 N.

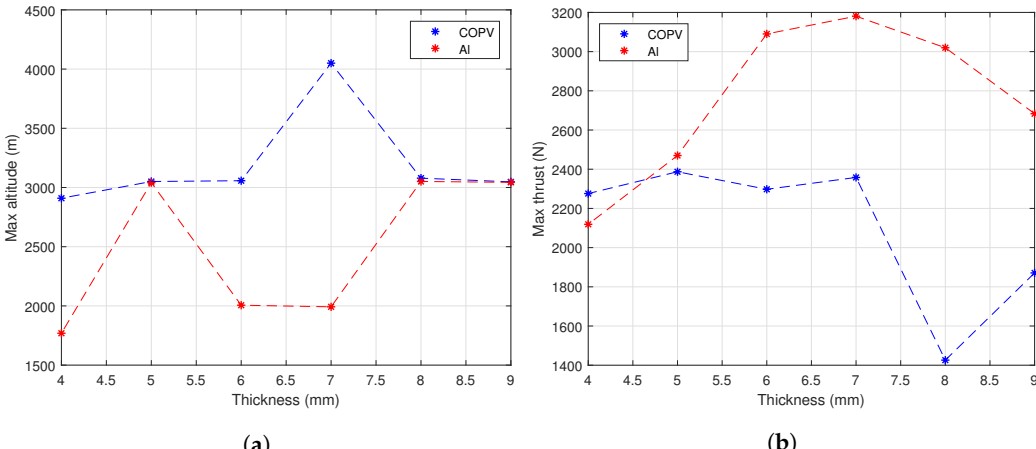

(**a**)  (**b**)

**Figure 18.** Maximum altitude (**a**) and thrust (**b**) achieved.

Taking into account an aluminium tank, the maximum altitude reached in Figure 18a is 3050.97 m with a thickness of 8 mm, while, for the COPV tank, the maximum altitude is 4050.03 m with a thickness of 7 mm. Considering the average maximum altitude, values of 2483.85 m and 3198.98 m are obtained for the aluminium and COPV tanks, respectively.

Based on these average results, it is possible to conclude that the tank design with COPV achieves the target altitude overcoming it, on average, by 198.98 m, which can be easily corrected by an air brakes system. On the other hand, the design with aluminium does not achieve the target altitude, although it is the one that yields the highest thrust values.

Figure 19a,b show the total rocket mass and the total oxidiser tank mass, respectively, after the optimisation process as functions of the thickness of the oxidiser tank. In Figure 19a, the highest rocket mass value for the case of the aluminium tank is 32.88 kg with a thickness of 5 mm and for the COPV tank 31.82 kg with a thickness of 7 mm. In terms of the average for the aluminium tank, the maximum rocket mass is 26.67 kg and for the COPV tank is 22.59 kg.

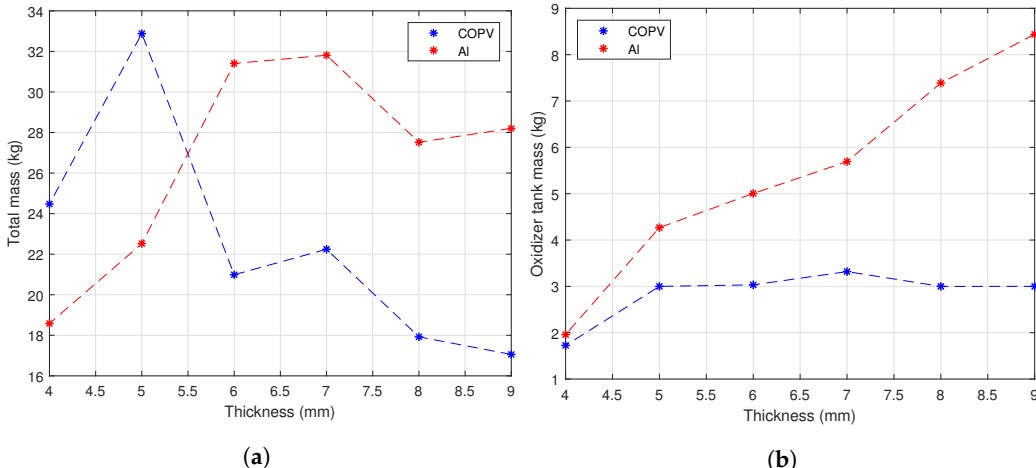

(**a**)  (**b**)

**Figure 19.** T otal mass of the rocket (**a**) and oxidiser mass (**b**).

The maximum value of the oxidiser tank mass made of aluminium is 8.43 kg, with a thickness of 9 mm, while its minimum value is minimum 1.95 kg, with a thickness of 4 mm. Regarding the COPV tank, its maximum value is 3.32 kg, with a thickness of 7 mm, and its minimum is 1.73 kg, with a thickness of 4 mm. In terms of average results, the aluminium oxidiser tank mass is 5.46 kg, while the COPV tank is 2.85 kg.

Based on these average results, it is possible to state that the tank design with a COPV achieves a total mass of 22.59 kg, while, regarding the design with aluminium, the mass reaches 26.67 kg, i.e., 4.08 kg heavier than the one with COPV, a reduction of about 15%. In all cases, the mass of the oxidiser tanks made of COPVs is lighter than with aluminium.

As a final remark, we may infer from Figures 18 and 19 that the launchers with oxidiser tanks made of COPV yield better performance in general: a higher apogee (Figure 18a) can be achieved with a lower total mass (Figure 19a) and consequently lower thrust (Figure 18b), which highlights the higher strength-to-weight ratio of COPVs. Therefore, this strongly motivates further research on the use of COPVs for these applications.

## 5. Concluding Remarks

A multidisciplinary design optimisation of a conceptual hybrid propulsion rocket system considering structural and aerodynamics disciplines has been presented. The design was carried out using an optimisation algorithm that considers the relevant process variables, the rocket mass, and the design of the oxidiser tank due to the direct relation between the size and weight of the tank, the dry mass of the tank, and the stored oxidiser mass.

The selected constraints were correlated with the limitations of the combustion chamber pressure to prevent combustion instability, ensure stability margin, meet geometric conditions, and ensure model consistency.

A conceptual design of the COPV was created using a finite element model to calculate the laminate stacking sequence, wall thickness, and dry weight, to be used as input in the MDO algorithm.

The use of the COPV to manufacture the oxidiser tank of a hybrid propulsion system led to an efficiency increase regarding the sounding rocket, as shown in the MDO process. This is evidenced by the lighter rocket designs obtained with COPV tanks that require less thrust for the same apogee when compared to those made of aluminium, thus emphasising their higher strength-to-weight ratio advantage.

The findings indicate that the problem studied is highly sensitive to the initial conditions, likely due to the quantity of design variables and the high level of constraints of the system. In such cases, small variations in initial conditions can lead to significantly different outcomes and solutions. This sensitivity highlights the importance of employing robust optimisation techniques and carefully selecting initial conditions to ensure the attainment of reliable and accurate results in the MDO process.

**Author Contributions:** Conceptualisation, A.S. (Alain Souza) and P.T.G.; methodology, A.S. (Alain Souza) and P.T.G.; software, A.S. (Alain Souza) and P.T.G.; validation, A.S. (Alain Souza) and P.T.G.; formal analysis, A.S. (Alain Souza) and P.T.G.; investigation, A.S. (Alain Souza), P.T.G., F.A., F.L., N.R. and A.S. (Afzal Suleman); writing—original draft preparation, A.S. (Alain Souza), P.T.G. and F.A.; writing—review and editing, F.A., F.L., N.R. and A.S. (Afzal Suleman); visualisation, A.S. (Alain Souza) and P.T.G.; supervision, F.A., F.L., N.R. and A.S. (Afzal Suleman); project administration, N.R. and A.S. (Afzal Suleman); funding acquisition, A.S. (Afzal Suleman). All authors have read and agreed to the published version of the manuscript.

**Funding:** This research was funded by Fundação para a Ciência e a Tecnologia (FCT), through IDMEC and INEGI, under LAETA, projects UIDB/50022/2020 and UIDP/50022/2020.

**Data Availability Statement:** The data presented in this study are available upon request.

**Acknowledgments:** Afzal Suleman acknowledges the NSERC Canada Research Chair Program.

**Conflicts of Interest:** The authors declare no conflict of interest.

## Abbreviations

The following abbreviations are used in this manuscript:

| | |
|---|---|
| MDO | Multidisciplinary Design Optimisation |
| COPV | Composite Overwrapped Pressure Vessel |
| HRPS | Hybrid Rocket System Propulsion |
| HRL | Hybrid Rocket Launchers |
| LAETA | Associated Laboratory for Energy, Transports and Aerospace |

|       |                                                   |
|-------|---------------------------------------------------|
| HRE   | Hybrid Rocket Engines                             |
| NIST  | National Institute for Standards and Technology   |
| CEA   | Chemical Equilibrium with Applications            |
| NASA  | National Aeronautics and Space Administration     |
| SAC   | Spaceport America Cup                             |
| CP    | Centre of Pressure                                |
| CG    | Centre of Gravity                                 |
| FEM   | Finite Element Method                             |
| IDF   | Individual Discipline Feasible                    |
| XDSM  | Extended Design Structure Matrix                  |

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
