# Peer review of "On the Multidisciplinary Design of a Hybrid Rocket Launcher with a Composite Overwrapped Pressure Vessel"

_jcs, doi:10.3390/jcs8030109_

Round 1
Reviewer 1 Report
Comments and Suggestions for Authors
The paper proposes a design approach for overwrapped composite pressure vessels that combines and synthetizes different disciplines and approaches. The study is interesting and helpful in defining a design methodology, but there are several aspects that need to be clarified and issues to be solved. Therefore, the manuscript requires a major revision before being reconsidered for publishing.
· Line 40 – Why did you consider the conventional configuration? Does it present more advantages than reverse and mixed hybrid configurations?
· Line 118-121 – Report a rationale for these hypotheses at the base of the analysis or mention a report for their justification.
· Sections 2.1, 2.2 and 2.3 – Are the analysis procedures reported in these sections original or conventional for this kind of propulsion system? Clarify this aspect and highlight the novelty points presented.
· Lines 193-194 – Could you briefly provide more details about the data set? What are the main features of rockets developed in this rocket competition?
· Line 248-249 – Give broader explanations about the thickness variation and fiber trajectories: report formulas and diagrams if available.
· Line 249 – How did you model the connection between the aluminum fitting and the composite material?
· Figure 9 – Using 2 linear elements in the thickness direction for the aluminum fitting can result in an incorrect simulation of the bending behavior. At least 3 elements are suggested for solid modeling in the thickness direction.
· Line 252-253 – What are the analyses and considerations behind this choice? Which failure criteria did you use for the assessment? Is this lay-up unsymmetric with respect to the middle plane? Give a proper rationale for this design solution and explain the design process to reach this configuration.
· Line 253 – Probably, it is better to replace “evaluate the stress levels in materials” with “relieve the stress levels in materials”.
· Figure 10 – Von Mises stress is inadequate to describe the stress state of composite materials. It is better to limit this contour plot to the aluminum fitting.
· Section 2.4 – Report also a figure with the displacement field of the tank.
· Line 265 – These safety margins are quite high. What impact of the other load conditions do you expect on the COPV?
· Add a link to the thesis reported as Ref. [29]. It cannot be found on the web. Whether such a link is not available, this reference should be removed.
· References [11] and [12] are inadequate for a scientific paper. Is any other source (book, conference procedia, paper etc.) available regarding these topics?
· Considering the amplitude and variety of scientific literature about the design of COPVs, the list of references should be enlarged, including more papers.
Author Response
Rebuttal [J. Compos. Sci.]: jcs-2883083
First of all, the authors want to express their gratitude for valuable time, comments, and questions raised by the reviewers to improve the quality of the current paper. All the changes performed to the manuscript are identified in the revised version in red. The authors’ comments and answers to the reviewers are presented in detail below.
REVIEWER 1
The paper proposes a design approach for overwrapped composite pressure vessels that combines and synthesizes different disciplines and approaches. The study is interesting and helpful in defining a design methodology, but there are several aspects that need to be clarified and issues to be solved. Therefore, the manuscript requires a major revision before being reconsidered for publication.
1. Line 40 – Why did you consider the conventional configuration? Does it present more advantages than reverse and mixed hybrid configurations?
ANSWER: We have chosen a conventional hybrid engine since it allows us to design the solid fuel (grain) by means of additive manufacturing with polymer based materials to reduce the carbon footprint.
2. Lines 118-121: Report a rationale for these hypotheses at the base of the analysis or mention a report for their justification.
ANSWER: We have included a recent reference to our MDO framework for hybrid rocket launchers (reference 32 in the revised version of the manuscript) where the assumptions made are described following the literature (references 33 and 34 in the revised version of the manuscript).
3. Sections 2.1, 2.2 and 2.3: Are the analysis procedures reported in these sections original or conventional for this kind of propulsion system? Clarify this aspect and highlight the novelty points presented.
ANSWER: These sections and subsections show the modelling process of each discipline applied to the MDO. The original part is in the integration of the structural design of the COPV in an MDO framework for the design of sounding rockets.
4. Lines 193-194: Could you briefly provide more details about the data set? What are the main features of rockets developed in this rocket competition?
ANSWER: The Data set is compose by some characteristic of a Rocket Engineering Competition. In these data are the real maximum altitude, the length, the diameter, the payload mass, the structural mass, the fuel mass, and the types of fuel and oxidizers. With this database, we were able to validate our simulation and optimization software.
5. Lines 248-249: Give broader explanations about the thickness variation and fibre trajectories: report formulas and diagrams if available.
ANSWER: A broader explanation of the tape trajectories modelling is added to the manuscript.
6. Line 249: How did you model the connection between the aluminium fitting and the composite material?
ANSWER: The connection between the aluminium fitting and the composite, was modelled using a cohesive interface with a stiffness K = 106. Interface failure was not considered in this analysis. The interface modelling details are added to the manuscript.
7. Figure 9: Using 2 linear elements in the thickness direction for the aluminum fitting can result in an incorrect simulation of the bending behavior. At least 3 elements are suggested for solid modeling in the thickness direction.
ANSWER: The figure was updated, although for the given application membrane stress due to pressure are more predominant.
8. Line 252-253: What are the analyses and considerations behind this choice? Which failure criteria did you use for the assessment? Is this lay-up unsymmetric with respect to the middle plane? Give a proper rationale for this design solution and explain the design process to reach this configuration.
ANSWER: A preliminary design of the COPV lay-up is performed using classic laminate theory using the first ply failure criteria and ensuring full coverage of the vessel dome. The laminate is defined in °æ layers to be adequate for filament winding or equivalent processes for axis-symmetric geometries. Details regarding the laminate modelling were added to the manuscript.
9. Line 253: Probably, it is better to replace “evaluate the stress levels in materials” with “relieve the stress levels in materials”.
ANSWER: The paragraph was improved.
10. Figure 10: Von Mises stress is inadequate to describe the stress state of composite materials. It is better to limit this contour plot to the aluminum fitting.
ANSWER: The figure was updated.
11. Section 2.4: Report also a figure with the displacement field of the tank.
ANSWER: Figure with displacements is also added to the manuscript.
12. Line 265: These safety margins are quite high. What impact of the other load conditions do you expect on the COPV?
ANSWER: Regarding linerless tanks, transverse failure corresponds to the weakest failure mode, that could lead to fluid leakage. Thermal strains and residual stresses could have severe impact on the effective transverse strength of the material, and could achieve near 50% of the load carrying capacity of the material. A statement regarding to the impact of other load conditions asides internal pressure was added to the manuscript.
13. Add a link to the thesis reported as Ref. [29]. It cannot be found on the web. Whether such a link is not available, this reference should be removed.
ANSWER: The reference [29] has been removed and the reference 10.3390/aerospace11020126 has been included. This uses the same database.
14. References [11] and [12] are inadequate for a scientific paper. Is any other source (book, conference procedia, paper etc.) available regarding these topics?
ANSWER: The references [11] and [12] have been removed.
15. Considering the amplitude and variety of scientific literature about the design of COPVs, the list of references should be enlarged, including more papers.
ANSWER: More references 19, 22, 23, 27 and 28 regarding COPVs design were added to the manuscript for completeness. However, this was not the primary focus of the present paper.
Reviewer 2 Report
Comments and Suggestions for Authors
This paper presents a multidisciplinary design optimization study of a hybrid rocket launcher concept, with a specific focus on exploring the benefits of using composite overwrapped pressure vessels (COPVs) for the oxidizer tank instead of conventional metallic vessels. The topic is interesting and has potential practical applications. However, some aspects of the paper need to be strengthened:
1. More background should be provided on prior work involving COPVs for aerospace oxidizer tanks and their demonstrated advantages to motivate the presented research. Quantify potential weight savings in some examples.
2. Details of the tank modeling using composites, including assumptions on materials/layup and failure criteria adopted, will make this section stronger from a materials viewpoint. Show key simulation results.
3. The optimization framework is discussed well, but more details are needed on the specific algorithms, constraints, variables, and objective functions chosen for this application.
4. Present clear quantitative comparisons between optimization outcomes for metallic and composite tanks in terms of rocket mass, altitude etc. The current results focus more on trends but specific numbers are needed.
5. The conclusions can further emphasize the major benefits found in using COPVs over metallic tanks based on the optimization study.
Comments on the Quality of English LanguageAdditional aspects like language, figures, and citations also need polishing.
Author Response
REVIEWER 2
This paper presents a multidisciplinary design optimization study of a hybrid rocket launcher concept, with a specific focus on exploring the benefits of using composite overwrapped pressure vessels (COPVs) for the oxidizer tank instead of conventional metallic vessels. The topic is interesting and has potential practical applications. However, some aspects of the paper need to be strengthened:
1. More background should be provided on prior work involving COPVs for aerospace oxidizer tanks and their demonstrated advantages to motivate the presented research. Quantify potential weight savings in some examples.
ANSWER: More references regarding COPV in aerospace applications were added (19, 22, 23, 27, 8)
2. Details of the tank modeling using composites, including assumptions on materials/layup and failure criteria adopted, will make this section stronger from a materials viewpoint. Show key simulation results.
ANSWER: Further details regarding, the layup definition and failure criteria were added.
3. The optimisation framework is discussed well, but more details are needed on the specific algorithms, constraints, variables, and objective functions chosen for this application.
ANSWER: The Multidisciplinary Design Optimisation is explained in Section 3 including the functional J(x). Constraints are defined in Section 3.2, and variables are shown in Table 3. Some additional remarks were added to the text.
4. Present clear quantitative comparisons between optimization outcomes for metallic and composite tanks in terms of rocket mass, altitude etc. The current results focus more on trends, but specific numbers are needed.
ANSWER: The advantages of using COPV is shown by the figures 18 & 19 and in the following paragraphs, which show a reduction in the total mass of the rocket in around 15% compared to the aluminium solution. The text was corrected to highlight this aspect.
5. The conclusions can further emphasize the major benefits found in using COPVs over metallic tanks based on the optimization study.
ANSWER: We have revised the manuscript’s conclusions to highlight the major benefits of using COPV for manufacturing the oxidiser tank of sounding rockets with hybrid propulsion.
Round 2
Reviewer 1 Report
Comments and Suggestions for Authors
The required amendments were considered and implemented.
The paper can be accepted in its present form.
Reviewer 2 Report
Comments and Suggestions for Authors
It has been revised accordingly.